# The Perception of Water Contamination and Risky Consumption in El Salvador from a Community Clinical Psychology Perspective

**DOI:** 10.3390/ijerph19031109

**Published:** 2022-01-19

**Authors:** Andrea Caputo, Manuela Tomai, Carlo Lai, Asia Desideri, Elpiniki Pomoni, Hilda Cecilia Méndez, Bartolo Atilio Castellanos, Federica La Longa, Massimo Crescimbene, Viviana Langher

**Affiliations:** 1Department of Dynamic and Clinical Psychology, and Health Studies, Sapienza University of Rome, 00185 Rome, Italy; manuela.tomai@uniroma1.it (M.T.); carlo.lai@uniroma1.it (C.L.); asia.sideres@gmail.com (A.D.); viviana.langher@uniroma1.it (V.L.); 2Institute of Geosciences and Earth Resources, National Research Council, 56127 Pisa, Italy; e.pomoni@igg.cnr.it; 3Faculty of Medicine, University of El Salvador (UES), San Salvador, El Salvador; hilda.mendez@ues.edu.sv (H.C.M.); bartolo.castellanos@ues.edu.sv (B.A.C.); claudia.cardoza@ues.edu.sv; 4National Institute of Geophysics and Volcanology, 00143 Rome, Italy; federica.lalonga@ingv.it (F.L.L.); massimo.crescimbene@ingv.it (M.C.)

**Keywords:** water, health risk, pollution, community clinical psychology

## Abstract

This study was carried out as part of the international cooperation project “Agua Futura” for the improvement of water resource management and the promotion of water, sanitation, and hygiene (WASH) behaviors in rural communities of Central America. Given the relevant healthcare problem of water pollution, especially in El Salvador, the aim was to detect some key factors affecting the perception of water contamination and its risky consumption from a community clinical psychological perspective. Ninety rural inhabitants of El Salvador were administered a structured interview exploring their perceptions about water quality and the impact of water consumption on health. Data were analyzed through a computer-aided thematic analysis—complemented by a qualitative analysis—allowing the detection of sense-making processes based on lexical variability. Different themes were identified with regard to the perception of water quality (i.e., mistrust, danger, and safety) and the beliefs about the impact of water consumption on health (i.e., rationalization, denial, awareness, displacement, and isolation of affect). The results showed heterogeneous perspectives about water quality and sanitation. However, the perceived negative impact of water consumption on health was mostly denied or minimized through massive distortions. Overall, the study highlights the role of defensive patterns in facing issues of water contamination, which may prevent the community from adopting healthy behaviors and adequate water management behaviors.

## 1. Introduction

In Central America, the reduced quality of drinking water represents one of the main health problems, especially in El Salvador [1,2], which shows the worst environmental conditions among the continental American countries. This is accompanied by the negative implications of climate change (e.g., droughts, floods, and landslides) since the average temperature of El Salvador has increased by 1.2 °C over the last 40 years. Indeed, according to Fondo Ambiental de El Salvador [3], El Salvador is the country of Latin America and the Caribbean with the lowest water availability per inhabitant, where the human right to drinking water and sanitation is not fully guaranteed. Only 32% of rural inhabitants have access to drinking water; in addition, 40% of water supply systems and 95% of rivers show high levels of pollution, with negative effects for the overall population’s health-related quality of life. The irregular provision of drinking water at a public level and the strong incidence of infectious diseases due to the contamination of drinking water make the sustainability of water supply systems the most relevant challenge for the country [4].

The World Health Organization has recognized the access to safe drinking water as one of the Millennium Development Goals’ targets [5]. Indeed, the lack of safe water supply systems is reported to be amongst the major causes of death in many low- and middle-income countries, due to both inadequate sanitation facilities and poor hygiene practices that may cause infectious diseases, including diarrhea, typhoid fever, polio, malaria, and schistosomiasis [6,7].

Along with increasing water scarcity, urbanization, and climate change, water pollution due to inadequate management of urban, industrial, and agricultural wastewater represents one the main challenges for public health. Despite the potential role of hygiene education and health promotion in reducing the incidence of infectious diseases in the general population, the effectiveness of such programs is quite limited [8,9,10], also in light of the strong internal motivation and effort required in adopting water, sanitation, and hygiene (WASH) behaviors as daily activities [11].

This is mostly intertwined with the reduced public environmental awareness and participation, which are considered crucial factors for successful pollution prevention [12]. Indeed, the public awareness of drinking water safety is relevant to both the promotion of household water treatment and the prevention of water contamination accidents [13,14,15]. Several studies attribute the ineffectiveness of WASH interventions to users’ difficulty in choosing safe water behaviors in daily household activities (e.g., chlorinated drinking water, adequate sanitation of drinking water containers) [16,17]. Engagement in preventive health behaviors seems to be not uniquely determined by the cognitive appraisal of objective environmental conditions, but is more widely shaped by health risk perceptions and judgments intended as intuitive and subjective interpretations of the world [18,19].

Therefore, a deeper investigation of personal meanings and perspectives is required for the understanding of apparently irrational decisions that deviate from a “scientific” approach to risk management [20]. When faced with environmental problems, individuals are not always receptive to information about healthy practices, regulations, or economic incentives [21], since their long-term behaviors are laden with implicit emotion [22,23,24]. As also stated by the Terror Management Theory (TMT), perceived environmental threats may evoke death thoughts and anxieties that are counteracted by cultural and psychological defenses, thus resulting in counterintuitive behaviors and decisions [25,26]. For instance, water-related risks could trigger automatic attempts to repress such death thoughts through minimization, displacement, rationalization, or sublimation as proximal and distal defenses that may also aggregate at a societal level in order to defend against vulnerability and manage future uncertainty [18]. Consistently, previous studies described emotions as predictors of behavior [27,28]. Positive and negative emotions, in fact, were found to intensify or attenuate environmental engagement and water consumption [22]. With regard to this, qualitative methods can be particularly fruitful in exploring symbolic aspects influencing risk perceptions and health behaviors regarding drinking water safety, so as to overcome most limitations of quantitative-based research [29,30]. Indeed, practices of water consumption and sharing are mostly shaped by collective sense-making processes at a community level [31], as also empirically supported by the link between community psychological factors (e.g., sense of belonging, mental well-being) and water contamination [11,32]. Then, the investigation of shared, locally grounded narratives may allow the identification of social and interpersonal dynamics, which may act as protective factors buffering against environmental risks from a psychosocial perspective [33]. Consistently, programs that develop water resources and WASH behaviors could benefit from incorporating the study of emotional and contextual variables that support change processes [34].

Based on these premises, the present research study aims at exploring the perception of water contamination and its risky consumption from the lived experiences of people belonging to rural communities that are daily faced with such issues. Specifically, it was carried out as part of the international cooperation project “Agua Futura” for the improvement of water resource management and the promotion of WASH behaviors in rural communities of Central America, especially El Salvador. This study adopts a community clinical psychological perspective, looking at the individual–community interface [35,36], so as to contribute to detecting some key factors potentially affecting shared perceptions and adjustment processes. Clinical community psychology is an emerging paradigm that integrates constructs, models, and tools from both community psychology and clinical psychology, predominantly with a psychodynamic orientation [35,36]. The integration effort stems primarily from the need to improve the effectiveness of psychosocial intervention in contexts of different breadth and complexity. Clinical community psychology focuses on the intersubjective emotional space cocreated by community members to ensure cohesiveness and belonging in facing anxiety feelings that arise from the uncertainty of the existence of community life [37,38]. In this regard, the enactment of mostly unconscious self-protective mechanisms is hypothesized as security operations that the community uses to defend against its painful realities or hide its real concerns about negative events such as environmental threats [39,40]. Therefore, the novelty of this qualitative study is the salience given to the community’s emotional life, which may help to construct rather than prove hypotheses regarding the perception of water-related issues. In this sense, it follows an action research paradigm where symbolic representations and implicit attitudes are taken into account to seek contextual knowledge and devise local solutions from a community clinical psychology perspective.

## 2. Materials and Methods

### 2.1. Setting

The present study was conducted in two pilot sites of San Salvador (i.e., San Marcos and Colima), which were identified on the basis of local needs and requirements, downstream of a shared and participated process between the Italian Agency for Cooperation and Development of El Salvador, project coordinators, and local actors (e.g., representatives of university institutions, ministerial bodies, administration officials, and community leaders). Along with the issues of water scarcity and pollution, such sites were characterized by over-exploitation of aquifers, soil change and loss of water control capacity, riverbed alterations, and degradation of wetlands (e.g., lakes, ponds, and estuaries). Drinking water services and sanitation were originally in charge of the municipalities; however, in 1961, the service was centralized through the creation of the National Administration of Aqueducts and Sewers (ANDA), which is an autonomous body funded also by private companies. Therefore, water resource services are not guaranteed at a public level but are mostly managed by local water committees, with charges that are not always sustainable by community members. This is especially true in rural areas, where most of the inhabitants have a monthly income of less than USD 300 and water is mostly drawn from private domestic wells, generally not subjected to sanitation procedures.

San Marcos is a municipality of El Salvador, located in a small and only partially urbanized valley, which is populated by low–middle-income families and has a local economy mostly based on itinerant trade. On the other hand, Colima is a canton of the municipality of Suchitoto, with an agricultural vocation and high archeological value, but also infrastructure deficiencies and scarce primary services. Both San Marcos and Colima represent two important natural areas characterized by a significant degree of biodiversity and crossed by several rivers, i.e., Malatapa, Cuapa, and Aguachía in San Marcos and Lempa, Acelhuate, and Los Limones in Colima. However, water resources have rapidly decreased over the last few decades because of urban growth and are strongly polluted (especially the Malatapa and Lempa rivers) due to human waste and agro-industrial residues [3]. Overall, drainage without wastewater treatment from the domestic dwellings represents 68% of the pollutant load, also because of the inadequate management of solid urban waste, ranging from 1504 to 3386 kg of BOD5/day (i.e., the five-day biochemical oxygen demand). Meanwhile, industrial pollution is mostly due to the use of agrochemical products in micro-basins and streams. This difficult situation is further aggravated by the risks associated with river overflow during the rainy season, the loss of large areas of secondary forest bands, as well as the lack of a strengthened risk and disaster prevention system and adequate training involving municipalities and communities. Health estimates are not available at a local level because of the deficiencies of healthcare facilities, especially in rural areas. Nevertheless, statistics about El Salvador show the relationship between the lack of access to safe water and the greater risk of illness and death, mainly including diarrhea and infectious (e.g., dengue and chikungunya) and respiratory diseases. The infant mortality rate in households without residential connections to potable water has been found to be approximately 40 deaths for each 1000 births [4].

### 2.2. Sampling and Recruitment

A convenience sample of 90 participants from the communities of San Marcos (20%) and Colima (80%) took part in the research study. The sampling was purposive and included both community inhabitants (92.2%) and community leaders (e.g., health promoters, representatives of ADESCO (Asociaciones de Desarrollo Comunal) and water management cooperatives, environmental guards) with the aim of ensuring enough diversity in roles, knowledge, and perspectives about water-related issues. Along with pragmatic reasons, the adequacy of the sample size was determined based on the saturation of data for computer-aided thematic analysis, intended as the degree of variation (rather than quantity) in the data [41]. Specifically, the type–token ratio (TTR; the ratio of different words or types to total words or tokens in the text) was employed as a measure of lexical variability and richness. This post hoc criterion was applied over the course of data collection to ensure a TTR less than 20% in the overall textual corpus, thus allowing a statistical approach to data [42]. The recruitment took place thanks to the logistical support in the field provided by the NGOs (non-governmental organizations), ACRA (Asociación de Cooperación Rural en África y América Latina) and ISCOS (Instituto de Cooperación Sindical en el Desarrollo). Participants were invited through door-to-door contacts facilitated by key informants with expertise and influence at the local level that made access to the community possible. Indeed, many areas of El Salvador are quite dangerous due to high homicide and crime rates, thus requiring the intermediation role of both NGOs and local leaders to ensure enough security to external visitors. Participants’ inclusion criteria were age of majority and sufficient language comprehension and production skills for participation in the administration procedure and informed consent provision.

### 2.3. Data Collection

Data for this study were collected through in-depth structured interviews, whose questions, as well as their wordings and order of administration, were determined in advance. The interviews consisted of open-ended questions overall aimed at facilitating associative processes about water-related issues within the community, so as to value participants’ lived experiences. A relevant topic covered by the interview guide related to perceived water quality and the beliefs about its impact on health in order to inspect participants’ awareness of water pollution. Specifically, two open-ended questions were formulated for this purpose as follows:What do you think about the water quality in the community? (question 1);What has the water consumption within the community entailed for your health? (question 2).

The interviews were conducted in pairs by Spanish native-speaking graduates enrolled in the specialization training course in Community Clinical Psychology of the University of El Salvador, respectively in the role of interviewer and observer, the latter with the task of audio-recording the interviews and annotating any relevant comments. The students received specific training on conducting the interviews and were supported by the teachers and researchers involved in the project. The interviews were mainly carried out in the homes of the interviewees, in a space that was as quiet and secluded as possible. The total interviews had an average duration of around 25 min (SD = 15). All participants were informed in advance about the study aims, and written informed consent was obtained before the interview administration and audio-recording. The interview was conducted in compliance with ethical principles and in accordance with the requirements of the Declaration of Helsinki and subsequent amendments. The study was approved by the Institutional Ethics Committee of the Department of Dynamic and Clinical Psychology, and Health Studies of “Sapienza” University of Rome.

### 2.4. Data Analysis

The collected interviews were transcribed verbatim by hand from the audio by a first researcher, and the transcriptions were checked against the original recordings for accuracy by another researcher. Thematic analysis [43] was adopted, facilitated by an automatic procedure conducted through the software program T-Lab PRO 4.1.1 [44]. Specifically, the “Context mining and automatic summary” tool was used, which provides an intelligent reading of the textual corpus and an initial summary of its contents. This procedure is particularly suitable for the exploratory purposes of a data-driven approach, where themes are directly derived from the data rather than from categories previously established by the researcher. Meaning is conceived as a property of word combinations and, depending on the “contextual effect”, based on co-occurrences of semantic traits (also named isotopies). Accordingly, word co-occurrence allows the detection of the syntagmatic relations between parts of the discourse and the consequent grouping of text segments (elementary context units, ECUs). This procedure enables the deconstruction of the typical structuring or ordered constituent parts of language, expressing rational and intentional contents, so as to focus on word selection during speech production, which reveals more implicit and affective associations towards the research object. Specifically, a digital “presence–absence” matrix with ECUs (i.e., text segments) in rows and lexical units (i.e., words) in columns is generated by the software. Then, an unsupervised clustering of the context units is automatically performed by using the bisecting K-means algorithm to obtain groupings of text segments sharing the same dictionary. The output consists of an HTML file presenting the whole corpus, with the selected ECUs highlighted in different colors based on the identified groupings.

This initial automatic summary was then complemented by a qualitative analysis of the text segments throughout the whole corpus to obtain an overall comprehension of the produced narratives, as well as to validate and potentially expand the identified themes through an iterative process. The identification of the initial themes for each open-ended question was independently performed by three researchers from the different groupings of text segments resulting from the previous cluster analysis. In this sense, the emerging themes were not based on a priori content categories. Rather, they were derived from the mapping of meanings directly emerging from the lexical variability characterizing participants’ interviews. Such themes were then systematized and synthesized by solving discrepancies by consensus. The entire analysis adopted an interpretivist approach and was supported by ongoing discussions within the research team and integrated the field notes to ensure enough data trustworthiness and triangulation among sources and perspectives. Specifically, the principles of Emotional Text Analysis (ETA) [45] were applied to grasp the symbolic level of texts, given the aim of detecting participants’ sense-making processes rather than their objective or factual knowledge. In ETA, the interpretation adopts a psychosocial perspective, stemming from constructivism and object relation theory as theoretical foundations, grounded in the motivational dynamics of social relationships (e.g., affiliation, power, achievement) connoting one’s emotional experience and orienting one’s relations with others and the context [46].

## 3. Results

Overall, 60% of the participants were female, with an age ranging from 18 to 83 years and a mean of 46.84 (*SD* = 17.05).

Thematic analysis allowed the identification of salient themes regarding the perception of water quality and the beliefs about the impact of water consumption on health. Such findings may provide an ecological frame for a better understanding of the awareness of water contamination and its risky consumption at a community level. This contributes to detecting potential dysfunctional attitudes and strategies enacted to mitigate health-related risks, which may prevent the community from adopting shared healthy behaviors, which can promote the adequate quality of water resources and ensure their effective sustainability in the long term. For each open-ended question, the emerging themes are presented below, followed by some examples of clusterized ECUs.

### 3.1. The Perception of Water Quality

Overall, three thematic domains have been identified regarding the perception of water quality, relating to feelings of mistrust, danger, and safety. The interviewees’ perspectives appear very heterogeneous and conflicting regarding the perceived harmfulness or healthiness of drinking water, as well as the effectiveness of precautions and sanitation procedures adopted at the community level.

#### 3.1.1. Mistrust

Water quality is perceived as problematic because of its doubtful potable standards that may make it unsuitable for human consumption. A general feeling of mistrust and uncertainty emerges also about the scarce activity of supervision of drinking water from local health units, thus leading community members to prefer purchasing bottled water, which is considered safer.
*“We do not consume this water although people claim that it is better than the bottled water we buy”*(Inhabitant, Woman, 42 years old)
*“To avoid getting sick it is better to buy packaged water, the health unit should supervise the actual quality of drinking water”*(Inhabitant, Man, 60 years old)

#### 3.1.2. Danger

Water is described as massively contaminated and harmful to health, triggering a constant sense of concern about water quality. Indeed, the sanitation procedures adopted at the community level are perceived as ineffective to ensure safe water and potentially dangerous because of the toxicity of chlorine used for disinfection.
*“Water is very contaminated here, chlorination just removes the dirt, the levels of chlorine should be monitored”*(Inhabitant, Man, 50 years old)
*“This water has a bad quality, it is not safe water, it is contaminated, we drink it only for pure necessity”*(Inhabitant, Woman, 65 years old)

#### 3.1.3. Safety

Water quality is perceived as adequate because of the filtration and sanitation treatments remediating its potential harmfulness, thus ensuring adequate safety levels. The extracts express trust in water sources and its direct consumption, which is described as a widespread practice among inhabitants.
*“The drinking water distributed in the community is suitable for consumption because the treatments provided through chlorine and filters make it of good quality”*(Leader, Woman, 47 years old)
*“Many people use the filter; see how beautiful this water looks, it comes out so crystalline that you can’t imagine”*(Inhabitant, Man, 52 years old)

### 3.2. The Beliefs about the Impact of Water Consumption on Health

The analysis identified five thematic domains with respect to the perceived impact of water consumption on health. Overall, only a small number of interviewees seemed to express enough awareness of risky consumption and the need for adopting protective health behaviors. Indeed, most of the participants resorted to several defensive strategies to handle perceived anxiety, through denying the problem of water contamination or mitigating personal vulnerability to the actual health-related risks.

#### 3.2.1. Rationalization

Interviewees acknowledged the poor quality of drinking water; however, the chlorination procedures, the use of filters at the taps, and past personal experiences were reported, which seemed to neutralize the perception of actual threats to human safety. A rationalization process is thus hypothesized, which aims at alleviating feelings of concern about water contamination through seemingly logical and consistent explanations.
*“We drank this water even in the past, but thank to chlorination nothing happened to us and I never heard that something happened to neighbors”*(Inhabitant, Woman, 75 years old)
*“Sometimes we have fish within the tanks that dies probably because of chlorine levels, but for human people it is different, we perceive nothing”*(Inhabitant, Woman, 42 years old)

#### 3.2.2. Denial

Water contamination and consequent health-related risks were substantially denied, through claims supporting the clarity of drinking water and its consequent suitability for human consumption. Interviewees considered the adopted precautions and disinfection procedures as unnecessary, refusing to admit a causal relationship between water quality and the possible onset of diseases, which were instead explained by resorting to religious beliefs.
*“Thank God, no problem. When we use water, we check if there is dirt or something similar, but it comes out so clear that there is no need to put a filter”*(Inhabitants, Man, 29 years old)
*“We have had no health problem, in truth people get sick because of the God’s providence that gives us the punishment we deserve”*(Inhabitant, Man, 73 years old)

#### 3.2.3. Awareness

Interviewees reported concrete and direct experiences of health problems associated with water consumption, especially intestinal tract infections and kidney diseases due to high levels of water contamination. Overall, there was a greater awareness of actual and serious risks, which led community members to enact safety and protection behaviors, particularly towards the community members considered to be most at risk, such as children.
*“This water has caused several health problems to many of us, such as urinary tract infections, since we all drink it”*(Inhabitant, Woman, 60 years old)
*“We use to boil the water taken from the well for our children, because kids are more likely to be affected by some disease”*(Inhabitant, Woman, 24 years old)

#### 3.2.4. Displacement

Interviewees did not report direct negative consequences of water consumption on their health, although they expressed some concern about the organoleptic qualities of water based on smell, taste, color, and turbidity. This may suggest a process of displacement, where anxiety feelings connected with water contamination are redirected to less threatening aspects, raising some doubts regarding the water’s suitability for human consumption.
*“Sometimes water may taste bad, but it probably depends on the use of chlorine”*(Inhabitant, Woman, 50 years old)
*“A little mold forms in the sink and, when the pipes are washed, water sometimes comes out so dark. This worries me”*(Inhabitant, Woman, 55 years old)

#### 3.2.5. Isolation of Affect

The problem of water contamination and its negative impact on health seem to be acknowledged only on a cognitive level. Interviewees appeared unconcerned since health problems were described with distantiation of time or place, thus suggesting a process of isolation of affect, where people avoid experiencing unpleasant feelings while remaining aware of the potential risks of water consumption.
*“There are a lot of people here with kidney failure but no tests have been carried out to confirm that it is caused by water”*(Leader, Man, 52 years old)
*“We know that sometimes water consumption may cause some stomach pain, vomiting or diarrhea; I think it’s just a matter of habit”*(Inhabitant, Woman, 50 years old)

## 4. Discussion

The present study aimed at detecting some key factors potentially affecting the perception of water contamination and its risky consumption by in-depth interviews with members (inhabitants and leaders) of rural communities characterized by environmental issues.

Overall, when examining their perspectives on water quality, participants express feelings of duplicitous nature. On one side, they mistrust and are worried due to problems of potential contamination and harmfulness of water sources; on the other side, they seem to show a self-reassuring attempt to emotionally outdistance the perilousness of their water. It has to be considered that water represents (and is) a vital resource. When the only available water is polluted, dangerous, and venomous, people have to inevitably face a deep anguish and fear from which they have to defend themselves.

A sense of unease may be particularly meaningful in those who cannot purchase bottled water, which is deemed safer, and are bound to consume tap water because of their poor economic possibilities. Indeed, people with lower socio-economic status are more likely to perceive water pollution as a problem because water quality is an issue of immediate concern due to their struggles to meet their basic needs [47,48]. Instead, risk perception is attenuated when participants consider precautions and sanitation procedures adopted at the community level as effective. In this regard, previous research has demonstrated that risk perceptions are negatively correlated with acceptance and benefit perceptions of sanitation systems [49], as well as with the general trust in the government’s capacity to manage water pollution through wastewater treatment [48]. It is thus possible that believing that sanitation procedures are effective contributes to alleviating anguish and fear of being poisoned by polluted water. In this sense, it can be considered a belief serving as a defense mechanism.

It should be noted that participants’ acknowledgment of environmental concerns is not directly associated with the perceived impact of water consumption on human health. Our study shows that awareness of water pollution among community members and the consequent need for safety protection behaviors is only partially present, in line with previous inconclusive findings about the link between environmental risk perception and preventive coping behaviors [50]. This result shows that comprehension-based awareness alone may not be enough to account for behaviors and attitudes towards polluted and contaminated water. Indeed, several defensive responses are enacted to counteract death anxieties triggered by perceived environmental threats according to Terror Management research [25,26]. Overall, four different defensive tactics are detected, regarding denial, displacement, rationalization, and isolation of affect. Consistently with a dual-process model of defenses [51,52], all the identified tactics seem to pertain to proximal defenses, which are more directly aimed at minimizing or suppressing death thoughts when they surface into consciousness, differently from distal defenses that keep death thoughts mostly unconscious through the maintenance of cultural worldviews and self-esteem. This suggests the strong anxiety feelings underlying participants’ narratives when faced with the issue of water pollution and its impact on health. Specifically, denial involves a refusal to accept that the problem of water contamination exists, and health-related problems are mostly explained as depending on God’s providence. This finding appears consistent with the protective role of religious beliefs in managing terror of death by affording a sense of psychological security and hope of immortality over environmental threats [53,54]. Another defensive response deals with displacement, which allows the marginalization of death thoughts, as found in previous studies, through distracting oneself from actual environmental risks [18,26]. Indeed, through displacement, participants move the discussion from the impact of water pollution on human health to sensorial information about the taste, color, and smell of the water. In such a way, they may redirect their anxieties to less threatening aspects of water quality that, albeit relevant to risk perceptions, trigger death thoughts to a lesser extent [48]. Moreover, two further mechanisms emerge, overall pertaining to intellectualizing defenses, namely rationalization and isolation of affect. As reported by previous studies [18,25,26], rationalization represents one of the main proximal defenses to protect against environmental problems. In this regard, rational defensive maneuvers reduce uncertainty feelings through cognitive and logical explanations, such as emphasizing the efficacy of water chlorination and filtered taps or reporting the lack of negative personal experiences in the past, so as to minimize one’s vulnerability. Meanwhile, isolation of affect allows community members to avoid death concerns while remaining aware of the unsafe water quality, mostly through pushing perceived health threats into a distant time or place [51,52], thus without considering water pollution as an emergency situation.

Based on the present findings, some practical implications can be derived for delivering locally grounded, community-based interventions for successful pollution prevention. First, the high heterogeneity of water quality perceptions suggests the lack of a common and unitary perspective on the issues of water contamination and related risky consumption among community members. Therefore, health education initiatives could be proposed in order to raise a shared awareness of actual water-related conditions by focusing on both cognitive and emotional aspects. This is especially important for those who currently perceive drinking water as safe and suitable for human consumption, since they could feel less vulnerable and exposed to environmental threats and risks. Along with this, low-income people as a target group should be particularly cared for in planning intervention activities, because of their significant psychosocial distress due to the reduced range of potential alternatives, such as purchasing bottled water, which they rely on to meet their basic needs. Indeed, it is also important to reinforce their sense of community and perceived support, so that increased awareness of water issues may not lead them to further feel helpless based on perceived disparities in water access [55,56].

Second, the enactment of defensive patterns to manage uncertainty and vulnerability in facing the impact of water consumption on health confirms the strong anxiety-evoking nature of water pollution discourse. From a community clinical psychological perspective, we should take into account the potential risk of failure of interventions primarily focused on water contamination because they could trigger counterphobic and avoidant attitudes, preventing participation in the offered programs. Indeed, as stated by Wolfe [26], proximal defenses could undermine individual motivation and engagement in water programs and even highlight societal incapacity to address systemic problems of water pollution, in turn worsening the underlying anxiety. In this regard, pollution-preventative initiatives should include a first step involving activities regarding the promotion of water conservation, so as to increase community perceptions of water as a resource to care for rather than as a threat to defend against. This could contribute to enhancing community members’ involvement and role in the possible resolution of water-related problems.

Therefore, the present findings seem to suggest the need for consideration of both technical (e.g., physical environment, sanitation procedures, hygiene information) and psychosocial variables (e.g., perceptual, affective, and relational aspects) when planning and delivering community-based interventions in the WASH field, by adopting a systemic and integrated perspective. In this regard, some potential risks should be carefully taken into account by policymakers in terms of practical implications. On the one hand, there is the risk of dealing with community members’ perceptions of water-related conditions exclusively from a cognitive perspective, assuming information provision as one of the main offered solutions. Indeed, this could represent a naïve and simplified vision of change processes, disregarding the emotional implications shaping the perceptions of the reality, attitudes, and practices. On the other hand, an exclusive focus on psychological variables should be avoided, as it does not consider the role of actual contextual factors as well as environmental, social, and economic resources. Indeed, this could lead to “blaming the victims” [56], as a tendency to consider disempowered local communities as being responsible for the social problems and deprived conditions that they experience.

Overall, some limitations should be acknowledged regarding the present study. The cross-sectional nature of the study design and the lack of a representative sample do not allow any generalization. Indeed, higher variability could characterize the experience of water quality and health-related risks; therefore, the found themes should be considered as preliminary and there is a need for further validation though more extensive studies or confirmation by public discussions at a community level. Other limitations refer to the lack of quantitative measures associated with interview data (e.g., perceived water pollution or health risk beliefs) and secondary sub-group analyses by participants’ characteristics (e.g., gender, age, socioeconomic status, etc.), which could provide more specific and accurate information.

This notwithstanding, the added value of the present study is the adoption of a data-driven approach that may provide culturally based knowledge, consistent with ecological and idiographic paradigms. Specifically, the use of qualitative data is better able to grasp specific sets of rules and symbolic meanings that can orient shared practices among community members, in order to plan tailored interventions and provide locally based solutions.

## 5. Conclusions

The preliminary conclusions of this study point out the relevance of inspecting symbolic representations and implicit attitudes towards water-related issues at a community level. Our findings show that, even in the face of an objectively identifiable problem such as that of water contamination, there is a wide range of subjective perceptions and beliefs about drinking water quality and its impact on health that can diversely orient human behavior. Specifically, when dealing with such a precious and vital resource as water, community members may not recognize the actual environmental threats they are exposed to, in order to defend against vulnerability and manage future uncertainty. This could inform policymakers in planning intervention strategies aimed at improving the management of water resources and promoting WASH behaviors and healthy practices within the community. Finally, we believe that knowledge of relevant aspects of community emotional life can contribute to the construction of shared rule systems that, as suggested by Ostrom [57], allow the local community’s self-management of collective natural resources.

## Data Availability

The data presented in this study are available on request from the corresponding author.

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
