# Peer review of "The Perception of Water Contamination and Risky Consumption in El Salvador from a Community Clinical Psychology Perspective"

_ijerph, 2022, doi:10.3390/ijerph19031109_

Round 1
Reviewer 1 Report
Thanks for the manuscript. This article focuses on an interesting question, but there are still many problems in the structure, organization and content analysis of the paper.
Structure
From the general structure and logic of the article, the research background should be at the forefront of the article.
Method
Qualitative research usually selects typical cases in the selection of research objects, rather than selecting representative samples like quantitative research. Therefore, the research should explain more about the method of selecting cases rather than samples. In addition, it should also explain why 90 cases were chosen instead of 100 or 80. According to the research criteria, the number of samples should be selected to meet the requirements of theoretical saturation. The research should inform readers how many cases meet the requirements of theoretical saturation and how they are reflected.
Results
1.The analysis texts and quoted utterances need to be clearly distinguished, and the quoted utterances should be in a different font from the main text.
2.In the part of water quality analysis, safety and danger are just the opposite, and they are actually two sides of the same thing and should not be regarded as a parallel relationship. At the same time, distrust and safety and danger are not on the same dimension. The authors need to re-construct the dimensions of water quality.
Moreover, the analysis of water quality lacks the necessary depth and is mainly based on quotes.
3.From the article title, the focus of the research is mainly on two points, one is the public’s perception of risk, and the other is coping behavior such as defensive Patterns Preventing Community Healthy Behaviors. While the authors point out that “ this article aimed at detecting some key factors potentially affecting the perception of water contamination and its risky consumption by in-depth interviews to members (inhabitants and leaders) of rural communities characterized by environmental issues”. Therefore, the content and title analyzed in this article are not completely consistent.
4.The authors propose that the impacts of water consumption on health (i.e. rationalization, denial, awareness, displacement, and isolation of affect), while the contents shows that they are not impacts of water consumption on health but their attitude toward the impacts of water consumption on health, therefore , the contents could not support the arguments.
On the whole, this is a rough paper, and it needs to be re-organized before being published
Author Response
Dear reviewer, thank you for your precious suggestions that we have taken into account for the revised version of the manuscript. As follows, our replies to each provided comment:
Structure
From the general structure and logic of the article, the research background should be at the forefront of the article.
R: As requested, the research background has been moved to the forefront of the article.
Method
Qualitative research usually selects typical cases in the selection of research objects, rather than selecting representative samples like quantitative research. Therefore, the research should explain more about the method of selecting cases rather than samples. In addition, it should also explain why 90 cases were chosen instead of 100 or 80. According to the research criteria, the number of samples should be selected to meet the requirements of theoretical saturation. The research should inform readers how many cases meet the requirements of theoretical saturation and how they are reflected.
R: The present study did not adopt a statistical sampling that can be considered representative of the entire communities involved. Participation was on a voluntary basis and the number of cases was determined from the saturation of data for computer-aided thematic analysis as better specified in the manuscript (lines 180-187)
Results
1.The analysis texts and quoted utterances need to be clearly distinguished, and the quoted utterances should be in a different font from the main text.
R: As suggested, the quoted utterances have been reported in italics with a smaller font.
2.In the part of water quality analysis, safety and danger are just the opposite, and they are actually two sides of the same thing and should not be regarded as a parallel relationship. At the same time, distrust and safety and danger are not on the same dimension. The authors need to re-construct the dimensions of water quality.
Moreover, the analysis of water quality lacks the necessary depth and is mainly based on quotes.
R: Further details about the method have been added (lines 237-266). The number of thematic domains were not established by the researchers, but emerged from the statistical procedure of the computerized text analysis (after the clusterization of text segments based on word co-ccurrences).
3.From the article title, the focus of the research is mainly on two points, one is the public’s perception of risk, and the other is coping behavior such as defensive Patterns Preventing Community Healthy Behaviors. While the authors point out that “ this article aimed at detecting some key factors potentially affecting the perception of water contamination and its risky consumption by in-depth interviews to members (inhabitants and leaders) of rural communities characterized by environmental issues”. Therefore, the content and title analyzed in this article are not” completely consistent.
R: As suggested, we have consistently changed the title into The Perception of Water Contamination and Risky Consumption in El Salvador from a Community Clinical Psychology Perspective”
4.The authors propose that the impacts of water consumption on health (i.e. rationalization, denial, awareness, displacement, and isolation of affect), while the contents shows that they are not impacts of water consumption on health but their attitude toward the impacts of water consumption on health, therefore , the contents could not support the arguments.
R: As suggested, we have accordingly changed the entire text. We now refer to “beliefs about the impact of water consumption”. This expression is more consistent with the administered open-ended question and better reflects the emerging distortions enacted to minimize risks.
Reviewer 2 Report
This is such a great topic to discuss and further discussion is demanded. Thus, high standard quality methods are needed to support this topic. Qualitative research is a suitable option to discuss this topic. But the data couldn't support the arguments and topic. The readers are not convinced by the interview contents which were listed in the paper. The authors mentioned some crucial actors, but it is difficult to persuade readers whether the people would tell the truth when there are others interests-related stakeholder in the same room.
The deep analysis on the two field sites are needed as well. The pollution situation, the disease statistics at local level, the interview of the doctors and so on.
Author Response
Dear reviewer, thank you for your precious suggestions that we have taken into account for the revised version of the manuscript. As follows, our replies to each provided comment:
This is such a great topic to discuss and further discussion is demanded. Thus, high standard quality methods are needed to support this topic. Qualitative research is a suitable option to discuss this topic. But the data couldn't support the arguments and topic. The readers are not convinced by the interview contents which were listed in the paper. The authors mentioned some crucial actors, but it is difficult to persuade readers whether the people would tell the truth when there are others interests-related stakeholder in the same room.
R: As suggested, we have further specified the method section in order to help readers understand the logical framework of data analysis, given the aim of detecting participants’ sense-making processes rather than their objective or factual knowledge. For clarity, others interests-related stakeholders that facilitated the access to communities were not present during the interview administration. We know that participants could be influenced by a variety of factors regarding the research context. Anyway, the computerized text analysis performed adopted a method focusing word selection during speech production, which reveals more implicit and affective associations towards the research object, apart from rational and intentional contents.
The deep analysis on the two field sites are needed as well. The pollution situation, the disease statistics at local level, the interview of the doctors and so on.
R: As suggested, we have included more information regarding the two field sites in both the “introduction” and the “setting” paragraph. We have clarified that health estimates are not available at a local level because of the deficiencies of healthcare facilities especially in rural areas. Therefore, disease statistics have been added that refer to El Salvador more widely.
Reviewer 3 Report
I would like to thank for the opportunity to review this paper.
The manuscript addresses one interesting topic and is very well structured. I really enjoyed reading it.
I have some comments that should be considered before this manuscript being accepted:
ABSTRACT
More information about Agua Futura (and the relevance of doing a qualitative study such as this one, so that the found results can have a context) and methods should be included.
INTRODUCTION
The novelty of the study should be better presented.
METHODS
Line 134 - Please indicate the full name of ADESCO
Lines 142-143 This information should be presented in the Results section since the main characterization of the participants is a result of the study
More information on how the potential participants were invited should be presented.
Ethical authorization information should be present.
DISCUSSION
The themes identified in the Results section should be better explored in the Discussion.
Strengths and limitations of the study should be addressed in the Discussion and not in the Conclusion.
CONCLUSION
A new Conclusion should be included in this section as most of the text regards to limitations of the study (which should be moved to the Discussion.
Author Response
Dear reviewer, thank you for your precious suggestions that we have taken into account for the revised version of the manuscript. As follows, our replies to each provided comment:
ABSTRACT
More information about Agua Futura (and the relevance of doing a qualitative study such as this one, so that the found results can have a context) and methods should be included.
R: As suggested, mote information about Agua Futura and the methods has been added.
INTRODUCTION
The novelty of the study should be better presented.
R: The novelty of the study has been expanded in the last part of the introduction.
METHODS
Line 134 - Please indicate the full name of ADESCO
R: The full name of ADESCO has been added.
Lines 142-143 This information should be presented in the Results section since the main characterization of the participants is a result of the study
R: As suggested, this information has been placed in the results section.
More information on how the potential participants were invited should be presented.
R: Further information have been added on how the participants were invited (in the paragraph sampling and recruitment)
Ethical authorization information should be present.
R: The ethical approval has been added.
DISCUSSION
The themes identified in the Results section should be better explored in the Discussion.
R: The discussion section has been expanded to better explore the identified themes.
Strengths and limitations of the study should be addressed in the Discussion and not in the Conclusion.
R: As suggested, the strengths and limitations of the study have been placed in the last part of the discussion section.
CONCLUSION
A new Conclusion should be included in this section as most of the text regards to limitations of the study (which should be moved to the Discussion.
R: As suggested, a new brief paragraph about the conclusions of the study has been added.
Round 2
Reviewer 2 Report
Dear Authors,
Thanks a lot for the review. The paper has been significantly improved to explain the research settings. You use a new title, and community is a research or interpret unit now.
If I understand it correctly, you don't offer a definition of the community perspective. A community is a group of people. If the paper is exploring public resources in a community. Maybe the works of Elinor Ostorm can help.
some related issues are: How rich are the people can avoid drinking pollution water? what are the major income of local households? How differences of livelihood in rich and poor families? will these different impact the community's emotion? what are influenced the most? How did people negotiate their public health issue or common pool resources in the history? was it successful? What are the memory of the community when the water was clean? and How do they express their ideas and emotion on behalf of a community rather than as an individual or a household
The relationship of the state's policy and local water management could be further analyzed. The role of the state/community will help the people understand why people are suffering.
Author Response
We would like to thank the reviewer for his/her useful suggestions. We think that now the manuscript shows further details that can improve readers’ understanding. Revisions are highlighted in red color.
Thanks a lot for the review. The paper has been significantly improved to explain the research settings. You use a new title, and community is a research or interpret unit now.
If I understand it correctly, you don't offer a definition of the community perspective. A community is a group of people. If the paper is exploring public resources in a community. Maybe the works of Elinor Ostorm can help.
R: We have added further details within the introduction about the community clinical psychological perspective we rely on. We have cited the Ostorm’s work in the conclusions to highlight the relevance of community self-management of natural resources.
some related issues are: How rich are the people can avoid drinking pollution water? what are the major income of local households? How differences of livelihood in rich and poor families? will these different impact the community's emotion? what are influenced the most? How did people negotiate their public health issue or common pool resources in the history? was it successful? What are the memory of the community when the water was clean? and How do they express their ideas and emotion on behalf of a community rather than as an individual or a household
The relationship of the state's policy and local water management could be further analyzed. The role of the state/community will help the people understand why people are suffering.
R: We have added further details in the setting paragraph to explain the state of the art of water resources services in El Salvador, especially in rural areas. As reported, community members are quite poor (mostly devoted to agriculture and itinerant trade activities) and cannot buy drinking water, which is drawn by private domestic wells, not subjected to sanitation procedures.